# The Effects of Integrated Step Training into the Physical Education Curriculum of Children with Intellectual Disabilities

**DOI:** 10.3390/ijerph182111340

**Published:** 2021-10-28

**Authors:** Pei-Fung Wu, Yu-Wei Chang, Tai-Been Chen, Li-Ching Chang

**Affiliations:** 1Department of Kinesiology, Health and Leisure Studies, National University of Kaohsiung, Kaohsiung 811726, Taiwan; zzz987z1031@gmail.com; 2Kaohsiung Municipal Nanzih Special School, Kaohsiung 811622, Taiwan; 3Department of Medical Imaging and Radiological Science, I-Shou University, Kaohsiung 824005, Taiwan; ctb@isu.edu.tw; 4School of Medicine for International Students, I-Shou University, Kaohsiung 824005, Taiwan; changlc@isu.edu.tw

**Keywords:** action research, step frequency, daily activities, self-confidence

## Abstract

(1) Background: This study investigated the changes in step frequency, walking ability, and standing posture of students with intellectual disabilities by integrating step training into the students’ physical education curriculum; (2) Methods: The centroid formula was used to estimate the geometric center of the students’ bodies in video footage of each participant. Each participant’s stepping frequency per minute was recorded. After training, the teachers involved were interviewed regarding the participants’ everyday activities in school. Each step training session was recorded by two video cameras. Each step training session was observed and photographed by a senior physical education teacher with special education qualifications; (3) Results: The step training increased the stability of the participants’ body axes. The participants’ average steps per minute of the participants significantly improved from 24.200 ± 7.554 to 28.700 ± 8.629. Additionally, despite the students exhibiting anxious behavior (e.g., squeezing their hands and grasping at their clothes) at baseline, the frequency of these behaviors decreased significantly from week 4. Overall, the students’ daily activities, motivation, interpersonal interaction, self-confidence, and anxiety behaviors improved; (4) Conclusions: After the 8-week step program, the participants with intellectual disabilities improved their step frequency, movement stability, ability to perform daily activities, walking speed, motivation, interpersonal interaction, and self-confidence, and they exhibited a lower level of anxiety-related behaviors.

## 1. Introduction

Intellectual disability is a disorder that manifests during childhood and is characterized by considerable limitations in both intellectual functioning and adaptive behavior, affecting numerous everyday social and practical skills [1,2]. Intellectual functioning refers to general mental capacity and encompasses learning, reasoning, and problem-solving. The perspectives of people with intellectual disabilities are increasingly being included in studies regarding their health. Unhealthy diet, sedentary behavior, and physical inactivity are the most common lifestyle problems among individuals with intellectual disabilities. Several studies have integrated the “healthy settings” approach into the daily routines of people with intellectual disabilities and trained support staff accordingly [3,4,5]. Moreover, the person-centered health model was recommended for people with intellectual disabilities [6].

Individuals with disability engage in less physical activity than their peers without disability [2,7]. For children, including those with disability, regular physical activity promotes social engagement and enhances body composition, bone health, and psychological health [8,9]. Although extensive research has examined the behaviors, patterns, and determinants of physical activity in children without disabilities, few studies have undertaken the crucial task of evaluating methods for improving physical activity in children with intellectual disabilities [10].

Children with intellectual disabilities have poorer physical health and development than the average student due to the limitations associated with the disability [11]. They are more likely to exhibit poor concentration, memory, imagination, thinking skills, and coping skills as well as delayed language development, self-centeredness, emotional instability, frustration, stubbornness, high help-seeking, low self-esteem, and self-defeating behaviors. When faced with a problem, children tend to learn how to solve the problem from adults or other peers, and after a successful experience, they repeat the same method. Generally, students with intellectual disabilities have delayed or deficient physical, psychological, and adaptive development, which in turn affects their performance in various aspects of their learning and social lives.

Shields and Synnot (2016) reported that children with mental and physical disabilities have lower physical strength and mobility than their peers (without disability) and that family support affects their participation in physical activities [12]. Howie et al. (2012) suggested that adults with intellectual disabilities, especially those not living in group settings, have fewer environmental resources and opportunities for physical activity and require interventions to increase their participation [13]. As individuals with intellectual disabilities have less experience with and fewer opportunities for physical activity than their counterparts without intellectual disabilities, improving their exercise habits, physical health, and life adaptability is crucial. For children with intellectual disabilities, refining school physical education programs is the optimal means of achieving these goals. Physical education programs focus on enhancing physical fitness, health, and sports-related skills and theories. In Taiwan, the special education curriculum emphasizes the general education curriculum as a basis and flexibly adjusts it to deepen, broaden, simplify, decompose, replace, and restructure curriculum elements according to the students’ abilities and needs [14]. Students with intellectual disabilities must learn a variety of sports in the physical education program. Therefore, despite the time constraints, teachers should consider how to enhance the physical activity level, physical activity motivation, and performance of students with intellectual disabilities through curriculum adaptation.

The World Health Organization (WHO) recommends for children and adolescents (aged 5–17 years) with or without intellectual disabilities to engage in an average of at least 60 min of primarily aerobic moderate-to-vigorous physical activity per day and to replace sedentary activity with physical activity of any (even light) intensity [15].

Step training, also known as step aerobics, is an aerobic exercise that can improve cardiorespiratory fitness [16], was popularized by US aerobics competitor Gin Miller in the 1990s. Step training programs are widely implemented in gyms, combining music rhythms and resistant exercises in a variety of exercises [17]. The 3-minute step test is used to assess aerobic fitness. Step training is often used in research studies focused on groups such as children, teenagers, and elderly people [18,19,20]. Most modern people maintain a physically inactive and sedentary lifestyle due to lack of time for exercise, which affects their health. Step training requires minimal space and is easy to learn and perform. Completing the recommended amount of daily exercise every day is sufficient to promote health and physical fitness. Until now, no studies have integrated step exercises into the physical education curriculum of children with intellectual disabilities. Therefore, this study incorporated step training into the physical education curriculum of students with intellectual disabilities to observe the effects on their step frequency, walking ability, and standing posture.

## 2. Materials and Methods

This study was conducted with the consent of participants and their parents or legal guardians, and parental consent forms were signed. The study was approved by the Human Research Ethics Review Committee of National Cheng Kung University (No. 106-036-2). The study participants comprised third-grade students in a special school in Kaohsiung City. Participants were identified as having an intellectual disability (Table 1). The study enrolled individuals who did not rely on walking aids. Those with major illnesses or mobility restrictions were excluded.

This study incorporated a step training into the physical education curriculum of a special school in Kaohsiung City. Excluding those with major illnesses or mobility restrictions, six male and four female students were initially recruited. Three of these participants had mild intellectual impairment (intelligence quotient ≤55 or mental quotient <70), six had moderate impairment (intelligence quotient ≤40 or mental quotient ≤54), and one had severe impairment (intelligence quotient ≤25 or mental quotient ≤39). The classification of intellectual disabilities in Taiwan is made according to the ICF-CY [21].

### 2.1. Study Design and Procedure

Step training was conducted twice weekly for 8 weeks as a component of the physical education curriculum, with two sessions unexpectedly suspended due to school activities (resulting in a total of 14 sessions). The protocol of the step training was based on the step-test protocol, which is based on the participants’ individual conditions with some modifications [22]. The initial speed was 50–80 beats per minute (BPM), and the stepper (Model 780, Taipei, Taiwan) height was set to 15 cm. Participants followed the teacher’s instructions to move their feet up and down in a sequential manner began with 1 min per session. During week 2, participants underwent 3-minute sessions, and the session durations were increased by 1 min per week thereafter. After 6 weeks, participants began performing 8 min of consecutive stepping to the rhythm of lively dancing music. The participants’ perceptions of the exercise intensity, which ranged from fairly light to somewhat difficult, were evaluated using Borg’s Rating of Perceived Exertion (RPE) scale [23]. The sequence of the step training comprised 3–5 min of stretching and warmup, the main session, and 3–5 min of stretching and cooldown. The students not participating in the program engaged in warmup and cooldown sessions and were in attendance to encourage the participants with cheers of support.

### 2.2. Data Collection

Each step training session was observed and photographed by a senior physical education teacher with special education qualifications. After each session, the supervising teacher wrote a diary entry describing whether the students were able to complete the step training, what the students’ emotional responses were, and what the teaching problems encountered were, if any. Six teachers were interviewed at the end of the 8-week training period to describe the physical and behavioral changes observed in students over that period. To avoid overlooking key data during the training sessions and provide a reference for analysis, review, and future teaching correction, two video cameras (PowerShot SX700 HS, Canon, Tokyo, Japan; HDR-PJ790V, Sony, Tokyo, Japan) were set up to record student movement during each step session. The participants’ number of steps per minute (during the exercise) were recorded before and after the completion of the step training program.

### 2.3. Extraction of Moving Participant Images from Video

The movement of each participant was recorded on two video cameras to indicate participants’ movement before and after the step training program completion on the same equipment. The movements of each participant were recorded for approximately 20 s by the same video camera. The configuration of these videos was as follows: 24 bits per pixel (i.e., RGB format), 30 frames per second, and 1080 p resolution.

### 2.4. Estimation of Participant Centroids from Extracted Images

First, videos, which contained approximately 600 frames (i.e., 30 frames/s × 20 s), were loaded into an algorithmic program. Each color frame (image) was converted into a grayscale image (Figure 1, Step 0 and 1). The background of the grayscale image was filtered out by applying the fast Fourier transformation algorithm with a radius of 50 pixels (Figure 1, Step 2 and 3). Then, this grayscale removed background image (RBI) was rebuilt using the inversed Fourier transformation (Figure 1, Step 4). Then, an averaging filter was applied on the RBI (ARBI) to obtain the shape of the participant. Next, the key parts of the participant were extracted from this ARBI output by using the *k*-means method with *k* equal to 4 (Figure 1, Step 5). A Canny edge detection approach was employed to depict the boundaries of participants after K-means processing (Figure 1, Step 6). The n points (*x*_1_, *y*_1_), …, and (*x*_n_, *y*_n_) are displayed in Figure 1, Step 6. The centroid formula was used to estimate the center of each participant as per Equations (1) and (2).
(1)$$C_{x}=\frac{1}{n}\overset{n}{\underset{i=1}{\sum }}x_{i}$$
(2)$$C_{y}=\frac{1}{n}\overset{n}{\underset{i=1}{\sum }}y_{i}$$

### 2.5. Statistical Analysis

The values derived from Equations (1) and (2), based on estimations of the center of the participants, are presented in terms of the mean ± standard error (SE). Stepping frequency data are presented in terms of the mean ± standard deviation (SD). A value of *p* < 0.05 (paired *t* test) was considered significant. All statistical analyses were performed using Sigmaplot, version 13.0 (Systat Software Inc., San Jose, CA, USA).

## 3. Results

Of the 10 participants, 5 had at least one or more recorded absences due to illness or other reasons. However, the participation rate of these five participants was still over 80%. The other five participants had no recorded absences.

### 3.1. Effects on Movement Posture during Stepping

After the step training, the stability of the participants’ body posture recorded during the step training, both in the horizontal and vertical axes, was significantly greater than that recorded before the training. As shown in Figure 2, the analysis revealed that for the five students who participated in the entire training, the amplitude of their body movement axes decreased significantly after the step training, both on the horizontal and vertical axes. Therefore, the step training can increase the stability of the body axis of the students with intellectual disabilities. As well as students with only mild intellectual disability, students with moderate and heavy intellectual disabilities exhibited significantly increased body movement stability during step training.

In addition, seven students increased their number of steps per minute through training. The students’ average number of steps per minute significantly improved from 24.200 ± 7.554 to 28.700 ± 8.629 (Figure 3). This increase in their step frequency may be attributed to the repetition of the movement practice because familiarity with the movement also reduced anxiety behaviors.

### 3.2. Effects on Standing Posture

In Table 2, student 1, or S1, exhibited uncoordinated backward leaning, stiff hands, shoulder shrugging in the early stages of training and later adopted a stiff and skewed standing posture, but achieved a straight posture without obvious anxiety movements such as shoulder shrugging or stiffness by the later stages of the program. S3 displayed an initial tension caused by hand incoordination (arm reversal), an unusually wide stance, and a tilted standing posture during the training process, but was able to relax with hands against the legs and feet together in an upright standing posture by the later stages of training. S5’s standing posture was stiff, with hand rubbing and squeezing in an anxious manner in the first training; with each week’s regular training, the standing posture gradually improved, becoming straight and no longer accompanied by hand rubbing or other anxious movements by week 7. S6 exhibited elbow flexion, slightly bent knees, and right-tilted standing posture in the early stages of training. With regular training, the knees could be straightened, the shoulder and neck muscles relaxed, and the standing posture straightened by the later stages of the program. S8 could not bring the feet together, grasped at clothing due to anxiety, and could not place the hands close to the thighs in the early stages of training. After gaining familiarity with the training, the feet could be brought together and the hands hung naturally against the legs.

### 3.3. Effects on the Performance of Daily Activities

Before the study began, a goal of 15 min was set for the maximum duration of the stepping in a single session, but after 8 weeks of training, the students went from being tired after 1 min to being able to sustain the exercise for 8 min. Although students with intellectual disabilities have innate limitations and special needs, with flexible adjustment of course objectives, sufficient practice time, and the use of multiple strategies, students can not only increase their motivation and sense of accomplishment but also achieve their course objectives.

After the 8-week step training program, we interviewed the six supervising teachers. Their assorted responses to the supplied questions were as follows.

Q1:How do you think the students’ overall performance of daily activities has changed since the step training?

“The students’ overall performance of daily activities has improved, and they are more focused and confident in their course work than before.”---------T1

“The students are less tired, and their daily activities are completed more efficiently.”---------T2

“The students have become more enthusiastic, and interaction among them has increased.”---------T3

“I feel that the students are more attentive in class, and they are more energetic than before.”---------T4

“During class, I heard S7 speak of losing weight, and I felt that S7 has become more confident.”---------T5

“The students’ movements have quickened and their performance of daily activities has improved greatly. S8 and S5 have become more confident.”---------T6

Q2:Do you notice any change in the students’ walking ability (ascending or descending the stairs or moving while carrying weight) after the step training?

“The students’ walking ability has improved and their movement speed has quickened.”---------T1

“The students are less short of breath and faster when they ascend or descend the stairs, which is an improvement.”---------T2

“The students can move more flexibly, and their walking ability has improved after the step training.”---------T3

“Yes, there has been an improvement in walking ability over the weeks of this semester.”---------T4

“An overall improvement in action capability is noticeable.”---------T5

“I feel that the walking ability of S8 has slightly improved.”---------T6

Q3:What activities do the students engage in after class? Have there been any changes since the step training?

“S8 spoke of being able to walk confidently and of a desire to take physical education classes.”Teacher’s Diary

“In the past, the students were less active in terms of class participation, but recently I found that students are very active in participating and assisting one another.”---------T1

“The students used to like to take the elevator, but now they can take the stairs by themselves and they walk very well.”---------T2

“A few students are brave enough to express their ideas in class, speak louder, and move faster.”---------T4

“S6 volunteered to help the teacher hand in something at the end of class, and spoke of being able to walk up the stairs from the first to the third floors without tiring, which impressed me greatly.”---------T6

Q4:What has happened to student participation in sports competitions? What are the changes?

“S7 spoke of being unafraid of joining the step training and wishing to participate in a stepping competition.”Teacher’s Diary

“Previously, when asked about the sporting competition, the students were hesitant to participate in it, but recently, I found that the students are willing to participate.”---------T1

“The students’ faces glow like never before when they win awards.”---------T2

“The spontaneous performance of the students during the sport competition was very impressive.”---------T3

“I heard S2 say, ‘I’ve already won 11 medals; I want to get the gold medal, so I’ll do my best in the competition.’”---------T5

Q5:How has the relationship between the students changed during the vocational training courses?

“In the past, the students barely interacted with one another in the vocational classes, but recently I have noticed an increase in such interactions.”---------T1

“S10 has more conversations with classmates and shows greater initiative in communicating with classmates.”---------T2

“Yes, the interaction between the students has increased, and the class atmosphere is lively, for example, S1 speaks louder and will take the initiative to talk to classmates.”---------T3

“S5 is more active in speaking and will actively call out to classmates.”---------T4

“It is the same as usual; I do not notice any change.”---------T5

“The students chat and joke with each other, interaction has increased, and peers get along well.”---------T6

## 4. Discussion

This study used qualitative data collection and quantitative analysis to investigate the physiological and psychological changes in students with intellectual disabilities after an 8-week physical education program. The amplitude of students’ body movement axes decreased significantly after completing the step training, both on the horizontal and vertical axes, indicating increases in movement stability (Figure 2) and step frequency (Figure 3). Interviews with teachers and observations of peer interactions among students revealed that the participating students exhibited obvious improvements in their walking ability, frequency of traversing stairs, level of motivation, peer relationships, self-confidence, and ability to engage in everyday activities after completing the step training program.

The WHO recommends for children and adolescents with or without intellectual disabilities to perform at least an average of 60 min of primarily aerobic, moderate-to-vigorous physical activity per day. However, achieving such a recommended level is difficult for children with intellectual disabilities, who are restricted in their development, tend to lack motivation, and tend to have passive lifestyles. The school-based physical education curriculum increases their opportunities to participate in daily physical activities. Although the participants in this study were only able to perform steps for 8 min at a time (failing to meet the initial goal of 15 min), after step training they exhibited improvements in their overall ability to engage in everyday activities, as reflected in their walking speed, participation in activities, interpersonal relationships, level of self-confidence, and level of motivation.

Step exercises do not require expensive equipment, and their intensity can be increased by raising the step height, increasing the cadence, or both [22]. Oliveira et al. (2016) suggested that 6-min walking and stepping tests may be viable alternative exercise prescriptions for patients with chronic heart failure [24]. In the present study, the participants exhibited significantly increased body movement stability (Figure 2) and an increased number of steps per minute (Figure 3) through repeated training, demonstrating the aerobic step program’s suitability for integration into physical education programs for children with intellectual disabilities.

For individuals with intellectual disabilities, participation in a healthy level of physical activity can not only maintain their physical health but also improve their quality of life, enhance their adaptability, and increase their employability and opportunities [8,25,26]. The physical education performance of students with intellectual disabilities is, in all respects, inferior to that of students without intellectual disabilities at all levels of education. Similarly, for individuals with intellectual disabilities, maintaining or improving upon their performance in the activities of daily life is difficult due to their various limitations, which may also affect their performance in the workplace [27]. As individuals with intellectual disabilities physically and mentally develop more slowly than those without intellectual disabilities, greater attention should be paid to their physical and mental health concerns [2,28]. Especially in adolescence, the lack of proper physical stimulation may inhibit the normal development of organs and tissues [29]. If proper exercise is performed during this period, the risk of many diseases, such as obesity, hypertension, and cardiovascular diseases, as well anxiety and emotional disorders can be greatly reduced [8,9,30,31]. Step training differs from general ball games, running, swimming, and other athletic sports, which necessitate skills, agility, and reaction abilities that require considerable time to learn. Step training is mainly composed of simple step movements, with fixed or variable rhythmic steps that can improve cardiorespiratory function, muscle strength, and other aspects of physical fitness [19]. The teaching process is student-centered, emphasizing the maintenance of a fixed frequency, step height, and training time to improve participants’ physical condition. Individuals with intellectual disabilities can achieve their exercise goals by following the instructions and performing rhythmic movements repeatedly. In the present study, after repeated step training, the participants increased their step frequency, stabilized their movement, and improved their ability to walk when engaging in everyday activity.

Each individual with intellectual disabilities responds differently to environmental stimuli due to their distinct physical and mental characteristics or level of prior learning [32]. For such individuals, even ostensibly simple events or activities require sufficient adaptation time due to changes in the environment or peer composition. In this study, even though the students were previously able to ascend and descend the stairs, the stepping exercise was a new activity for them. Hence, the students displayed some anxiety-induced behaviors, such as being easily distracted, grabbing at their pants, straining their hands, and stiffening their body movements. Forte et al. (2011) noted that students with intellectual disabilities are more likely to have anxiety than their peers without such disabilities, suggesting that assessing the problem behaviors of students with intellectual disabilities could be a means of assessing anxiety [33].

Modern technological advances, automated equipment, and computer-aided living have reduced opportunities for physical activity, and human lifestyles have gradually become more sedentary, which represents a health concern for the general public as well as for people with intellectual disabilities [34]. This lack of physical activity also impedes the development of large and small muscle groups. Therefore, due to the impairment and limitations of their congenital conditions, people with intellectual disabilities are not only affected by inferior physical performance, muscle development, and activity levels, but also poor self-esteem and self-confidence [5]. In this study, the dynamic training program led to more interaction, conversation and encouragement among students with intellectual disabilities as well as a positive and enthusiastic classroom atmosphere. Interviews with teachers and observations of peer interactions among participants, exhibited improvement in their walking ability, frequency of traversing stairs, level of motivation, peer relationships, self-confidence, and ability to engage in everyday activities after 8 weeks of the step training program. In conclusion, the step training program yielded positive effects for all participants with intellectual disabilities at both the physical and psychological levels.

This study had some limitations. First, the scheduling of the sessions was influenced by school celebrations or holidays and therefore could not be conducted for 8 consecutive weeks. Second, the Ministry of Education’s curriculum guidelines governing 12-year basic education in health and physical education disallow step training (cardiorespiratory endurance) to be included in all 20 weeks of a school semester. This study was primarily a qualitative one. Through video and photographic records, we observed how the students moved during the stepping process, and teachers were interviewed regarding changes in how students participated in everyday school activities after completing the training. To obtain more quantitative data, further studies could use a metronome to control the stepping speed in programs involving participants with intellectual disabilities and a stable gait.

## 5. Conclusions

Repeated engagement in step training increases movement stability and therefore stepping frequency; it also improves walking speed, motivation, interpersonal interactions, self-confidence, overall ability in engaging in everyday activities. Step training should be incorporated into the physical education curriculum for students with intellectual disabilities.

## Figures and Tables

**Figure 1 ijerph-18-11340-f001:**
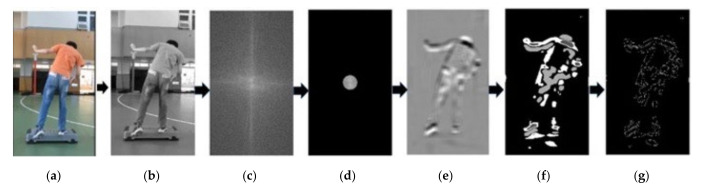
The steps of estimated centroid from an RGB image. (**a**,**b**) The image was converted into a grayscale image; (**b**,**c**) The background of the grayscale image was filtered out by applying the fast Fourier transformation algorithm with a radius of 50 pixels; (**d**) The grayscale removed background image (RBI) was rebuilt using the inversed Fourier transformation; (**e**) An averaging filter was applied on the RBI (ARBI) to obtain the shape of the participant; (**f**) Next, the key parts of the participant were extracted from this ARBI output by using the *k*-means method with *k* equal to e; (**g**) A Canny edge detection approach was employed to depict the boundaries of participants after K-means processing.

**Figure 2 ijerph-18-11340-f002:**
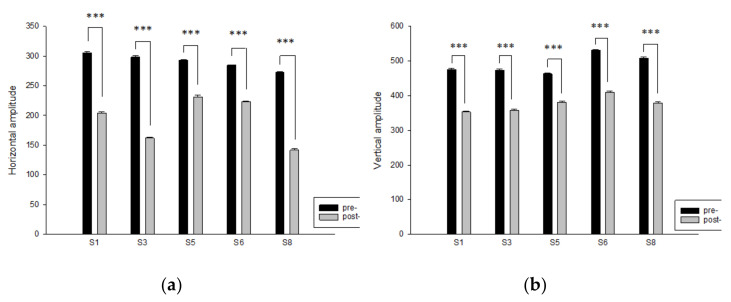
Step training increased the body movement stability of the students with intellectual disabilities. (**a**) Data were obtained using Equation (1), $C_{x}=\frac{1}{n}\overset{n}{\underset{i=1}{\sum }}x_{i}$; (**b**) Data were obtained using Equation (2), $C_{y}=\frac{1}{n}\overset{n}{\underset{i=1}{\sum }}y_{i}.\;$ Data are presented as means ± standard errors (SE) based on estimations of the center of the participants. A value of *p* < 0.05 (paired *t*-test) was considered significant. *** < 0.001.

**Figure 3 ijerph-18-11340-f003:**
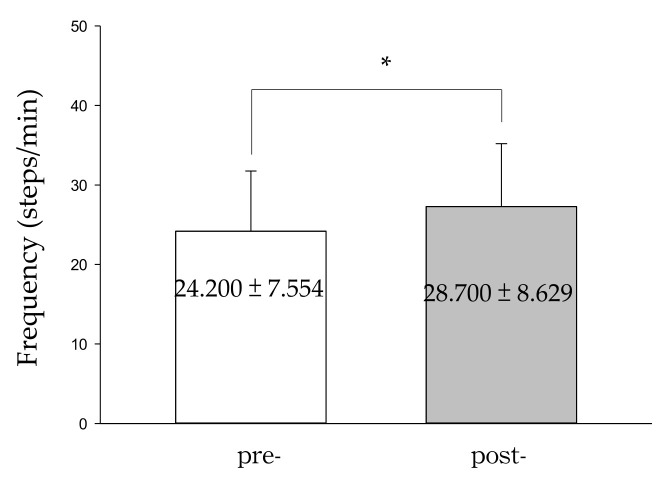
Stepping training improvement the step frequency of students with intellectual disabilities. Data were presented as mean ± standard deviation (SD) collected from the number of steps per minute before and after stepping training. A value of *p* < 0.05 (paired *t*-test) was considered significant. * < 0.05.

**Table 1 ijerph-18-11340-t001:** Participant information.

	Gener	Age (Years)	Disability Level	Mobility
S1	M	18	Moderate	Physically weak, able to walk and ascend and descend stairs unaided.
S2	M	19	Mild	Good physical performance, good coordination, and able to walk fast and ascend and descend the stairs.
S3	M	18	Mild	Physical performance is fair, can walk alone and ascend and descend the stairs, but body movements are slightly stiff and uncoordinated.
S4	M	18	Moderate	Weak physical ability, can walk alone and ascend and descend the stairs, but exhibiting uncoordinated body movements.
S5	M	18	Heavy	Physical performance is fair, can walk alone, must hold the handrail to ascend and descend the stairs, uncoordinated body movements.
S6	M	18	Moderate	Physical performance is slightly weak, with the ability to walk and ascend and descend the stairs, but the body movements appear stiff and uncoordinated.
S7	F	18	Mild	Physical performance is weak, can walk and ascend and descend the stairs, but body movements are stiff and poorly coordinated.
S8	F	19	Moderate	Physical performance is fair, with the ability to walk, and the need to adhere to the handrail when going up and down the stairs, with slightly stiff body movements and lack of coordination.
S9	F	19	Moderate	Fair physical performance, good physical coordination, good walking ability, and ability to ascend and descend the stairs.
S10	F	18	Moderate	Body type is obese, and physical performance is obviously weaker, with the ability to walk. Can ascend and descend the stairs alone, but coordination of movement is lacking.

**Table 2 ijerph-18-11340-t002:** Standing posture changes.

	Standing Posture Changes	
	Before	After
S1	During the pretraining period, the body leaned backward indicating incoordination, with stiff hands, shoulder shrugging, and a stiff and crooked standing posture.	After training, the standing posture was straight and anxious movements such as shoulder shrugging and stiffness were absent.
S3	Initially, tension caused hand incoordination (ectropion) and inability to bring the feet together, resulting in a tilted body.	After training, the hands were relaxed, the hands hung down naturally, the feet were together, and an upright standing posture was adopted.
S5	In the first week, the standing posture appeared stiff, with rubbing and straining of the hands, and an anxious appearance.	With regular weekly training, the standing posture gradually improved, and by week 7, a straight standing posture could be adopted without any appearance of anxiousness (e.g., rubbing hands).
S6	Elbows were bent and knees bent slightly in the pretraining period, with body tilted to the right in the standing position.	After regular weekly training, the knees were straight, the shoulder and neck muscles were relaxed, and the standing posture was no longer skewed.
S8	Initially, the feet could not be brought together, hands clutched at the pants because of anxiety, and the hands could not remain at the thighs.	At the end of training, the feet could be placed together and the hands hung down naturally.

## Data Availability

Photos or videos that involve the privacy of the participants are not available for request.

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
