# Peer review of "The Effects of Integrated Step Training into the Physical Education Curriculum of Children with Intellectual Disabilities"

_ijerph, 2021, doi:10.3390/ijerph182111340_

Round 1
Reviewer 1 Report
I wish to commend the authors on completing this body of research. This is an area of study that requires more attention and it is welcomed to see research progressing within this area. In general, the research is interesting. In my review I hope to provide kind and helpful feedback in order to add to the robustness of this publication. I include below some queries and comments which should be addressed and considered by the authors:
-The title of the article should be specific and read more concisely e.g. The Effects of Integrating Step Exercise to music into the Physical Education Curriculum of Children with Intellectual Disabilities
-Line 20- state physical ability…is it physical fitness or ability you assessed? Your outcome measures and methods of assessing the outcome measures much be presented in a valid, reliable and accurate manner- for example physical fitness and physical ability are both different parameters and measured differently.
-The abstract should state what outcome measures were analysed. There is also no reference to statistical findings?
-Keywords provided should be different from the title- please remember accuracy, reliability and validity of terms used e.g. physical ability vs fitness
-Line 12 and 13: physical fitness, walking ability, and standing posture- how were these assessed… measured using teacher diaries, interviews with relevant teachers, classroom videos, and observation records- can you assess physical fitness via these methods?
-Line13: states ‘aerobic step training’- be consistent…add ‘aerobic’ to title also?
-line 32 The disability originates before the age of 18 years (Bertelli, et al., 2016; Schalock et 32 al., 2010)…This sentence does not make sense-is this meant to form part of a sentence?
-line 49/58 : references are required for this paragraph e.g. Children with intellectual disabilities have poorer physical health and development than the average student due to the limitations associated with the disability-provide references
-Line 77: The authors use the term ‘Step aerobics’ be consistent in your intervention name throughout the document e.g. step exercise/step training/step aerobics etc- be consistent
-In the introduction reference should be made to the WHO physical activity guidelines for health for individuals with disabilities. https://www.who.int/news-room/fact-sheets/detail/physical-activity
-Line 105/6: … intensity of the regular step training was adjusted according to the students' individual conditions…this needs expanded. How was this done and why? Based on what guidelines? %MHR? Your methods/intervention should be able to be repeated by a blinded researcher- as it currently reads this would not be possible.
- Was the stepping exercise completed in the presence of music as was alluded in the introduction? Or in silence? If in time to music what type of music? Random music selection/upbeat songs/etc etc
-Reference should be made to the reasons for only five participants completing the intervention- why did the others drop out? Was a sample size calculation completed?
-Line 110: font size is different
-Line 112-13, do you mean low to moderate intensity?
-Line 116/17: a diary entry describing student movement performance, student emotional responses, and teaching problems encountered- Can you provide a template of this, for example what type of descriptions were provided for ‘movement performance’? Was there a scale provided? How were each of these outcomes defined?
-Diary entry, teacher interviews and Video recordings- in the abstract in addition to these three methods of recording data you also state ‘observational records’? Is this the same as the diary entrys? Consistency and clarity is required throughout the document
-Line 161-164: This is a sweeping statement- you should provide the data or refer to it in a past/future publication or remove these lines. Why is the data available for seven participants if only five completed the intervention?
-Detail must be added to the methods section outlining how each of the outcome measures were collected e.g. in the results section you refer to body posture analysis- how would a blinded researcher replicate these assessments? The details must be provided in the methods.
-It may also be helpful to include how you defined mild, moderate and heavy disability within the methods section.
-Throughout the results section you must provide the figures from the statistical analysis in the text- you state statistical significance was found e.g. significantly greater- provide the p values etc at on these occasions.
-Line 86/7: had students reached volatile exhaustion after 1 min? Did they choose to stop or did the assessor stop them? More detail required
-Line 184- the effects on physical performance – I am unclear as to how ‘physical performance’ is assessed. Firstly, your terminology around physical fitness/performance etc must be refined and consistent as discussed previously. Are you simply measuring time or duration of stepping? This is very different to measuring an aspect of physical fitness such as aerobic capacity? Please refine and define this throughout the document
-Line 187-191- I am unsure if this is appropriate here? Reads like an opinion?
-Figures and tables should be provided where they are first mentioned in text and consistent with the publishers guidelines
-Line 278- ‘frequency of climbing’ is introduced as another outcome measure. This is difficult to follow, you must be consistent when presenting your outcome measures and their results. You also mention physiological and psychological changes as outcome measures above here for the first time??
-Line 279-281 requires a reference
-Check font size and colour throughout the document- there are inconsistencies
-Line 279-301- this section reads like a list of facts? What is this adding to the discussion of your research findings? It seems as if it may have been copied and pasted from a literature review of the area- I am unsure if it fits here. Would using the WHO physical activity guidelines and providing some statistics on participation of individuals with disability add more to this section and allow the discussion to flow into the next paragraph?
-Line 313-14: Especially in adolescence, the lack of proper 313 physical stimulation may inhibit the normal development of organs and tissues …requires a reference
-Line 338-44 requires references
-Line 348/9: You state… demonstrated improvement in their anxiety behavior, interpersonal relationships, and self-confidence after 8 weeks of the step training program…how can you conclude this? Make conclusions based on what your outcome measures where and your findings…for example how did you assess participant self-confidence? You subjectively asked teachers on their opinion- your conclusions must reflect your findings.
-Line 351/55: this needs rewording.
-Line 358- in your conclusions you mention music tracks for the first time- this detail must be provided in your methods section.
-Lines 359-61: This is the first time that encouragement from peers was discussed- this should be in your methods section.
-There should be no new information in your conclusion section- your conclusions should simply conclude and summarise your studies findings. The conclusion section should be revised.
-Limitations of the study are not presented. This should be completed.
-Future research is not presented or explored. This should be provided.
-Line 324-26- is there an argument for using walking to music within this population? It is simple, cost effective, accessible etc- music provides motivation and studies have shown that it allows regulation of cadence to tempo. The additional aspect of walking in nature could add another potential for future research and be in line with WHO and ISPAH physical activity targets
Author Response
Response to Reviewer 1
I wish to commend the authors on completing this body of research. This is an area of study that requires more attention and it is welcomed to see research progressing within this area. In general, the research is interesting. In my review I hope to provide kind and helpful feedback in order to add to the robustness of this publication. I include below some queries and comments which should be addressed and considered by the authors:
-The title of the article should be specific and read more concisely e.g. The Effects of Integrating Step Exercise to music into the Physical Education Curriculum of Children with Intellectual Disabilities
Author response: We have revised the manuscript title. “The Effects of Integrated Step Training into the Physical Education Curriculum of Children with Intellectual Disabilities”
-Line 20- state physical ability…is it physical fitness or ability you assessed? Your outcome measures and methods of assessing the outcome measures much be presented in a valid, reliable and accurate manner- for example physical fitness and physical ability are both different parameters and measured differently.
Author response: We thank the reviewer for his/her reminders. We have revised it (line 13, 26-29 highlighted in blue, abstract section).
-The abstract should state what outcome measures were analysed. There is also no reference to statistical findings?
Author response: We have rephrased the abstract section (line 20-29, highlighted in blue).
-Keywords provided should be different from the title- please remember accuracy, reliability and validity of terms used e.g. physical ability vs fitness
Author response: We have revised the keywords (line 30, highlighted in blue).
-Line 12 and 13: physical fitness, walking ability, and standing posture- how were these assessed… measured using teacher diaries, interviews with relevant teachers, classroom videos, and observation records- can you assess physical fitness via these methods?
Author response: Some of the text is not very relevant and we have revised.
The amplitude of students’ body movement axes decreased significantly after completing the step training, both on the horizontal and vertical axes, indicating increases in movement stability (Figure 2) and step frequency (Figure 3) (line 261-261). The students’ average of the number of steps per minute significantly improved from 24.200 ± 7.554 to 28.700 ± 8.629 (Figure 3). In the present study, after repeated step training, the participants increased their step frequency, stabilized their movement, and improved their ability to walk when engaging in everyday activity (line 298-299).
For a more complete presentation of the results, the results of the step frequency per minute before and after step training are showing in Figure 3.
-Line13: states ‘aerobic step training’- be consistent…add ‘aerobic’ to title also?
Author response: We have revised it. We use “step training” in the revised manuscript.
-line 32 The disability originates before the age of 18 years (Bertelli, et al., 2016; Schalock et 32 al., 2010)…This sentence does not make sense-is this meant to form part of a sentence?
Author response: We have rephrased (line 33-35).
-line 49/58 : references are required for this paragraph e.g. Children with intellectual disabilities have poorer physical health and development than the average student due to the limitations associated with the disability-provide references
Author response: We have added the reference, [11] (line 50).
Ref. 11. Eisenhower, A. S.; Baker, B. L.; & Blacher, J. Preschool children with intellectual disability: syndrome specificity, behaviour problems, and maternal well-being. Journal of Intellectual Disability Research 2005, 49, 657–671. https://doi.org/10.1111/j.1365-2788.2005.00699.x
-Line 77: The authors use the term ‘Step aerobics’ be consistent in your intervention name throughout the document e.g. step exercise/step training/step aerobics etc- be consistent
Author response: We have revised. “Step training” was used in the revised manuscript.
-In the introduction reference should be made to the WHO physical activity guidelines for health for individuals with disabilities. https://www.who.int/news-room/fact-sheets/detail/physical-activity
Author response: We have added the WHO guideline and rephrased it (line 71-74).
The World Health Organization (WHO) recommends for children and adolescents (aged 5–17 years) with or without intellectual disabilities to engage in an average of at least 60 minutes of primarily aerobic moderate-to-vigorous physical activity per day and to replace sedentary activity with physical activity of any (even light) intensity [15] (line 71-74).
Ref. 15. Physical activity. https://www.who.int/news-room/fact-sheets/detail/physical-activity (26 November 2020)
-Line 105/6: … intensity of the regular step training was adjusted according to the students' individual conditions…this needs expanded. How was this done and why? Based on what guidelines? %MHR? Your methods/intervention should be able to be repeated by a blinded researcher- as it currently reads this would not be possible.
Author response: We have rephrased it in material and methods section (line 101-104, 105-109). We also added two references.
The protocol of the step training was based on the step-test protocol, which is based on the participants' individual conditions with some modifications [22]. The initial speed was 50–80 beats per minute (BPM), and the stepper (Model 780, Taipei, Taiwan) height was set to 15 cm (line 101-104).
During week 2, participants underwent 3-minute sessions, and the session durations were increased by 1 minute per week thereafter. After 6 weeks, participants began performing 8 minutes of consecutive stepping to the rhythm of lively dancing music. The participants’ perceptions of the exercise intensity, which ranged from fairly light to somewhat difficult, were evaluated using Borg's Rating of Perceived Exertion (RPE) scale [23] (line 105-109).
Ref. 22. Howley, E.T.; & Franks, B.D. Health/Fitness Instructor’s Handbook. Champaign, IL: Human Kinetics Publishers, Inc., 1986.
Ref. 23. Borg, G.A. Psychophysical bases of perceived exertion. Medicine & Science in Sports & Exercise 1982, 14, 377-381.
- Was the stepping exercise completed in the presence of music as was alluded in the introduction? Or in silence? If in time to music what type of music? Random music selection/upbeat songs/etc etc
Author response: Actually, after 6 weeks, participants began performing 8 minutes of consecutive stepping to the rhythm of lively dancing music (line 106-107). We have rephrased it in material and methods section.
-Reference should be made to the reasons for only five participants completing the intervention- why did the others drop out? Was a sample size calculation completed?
Author response: Of the 10 participants, 5 had at least one or more recorded absences due to illness or other reasons. However, the participation rate of these five participants was still over 80%. The other five participants had no recorded absences. We have rephrased it in results section (line 156-158).
-Line 110: font size is different
Author response: We have revised it.
-Line 112-13, do you mean low to moderate intensity?
Author response: We have revised it. We have added the intensity of stepping in the material and methods section (line 101-104, 105-109).
-Line 116/17: a diary entry describing student movement performance, student emotional responses, and teaching problems encountered- Can you provide a template of this, for example what type of descriptions were provided for ‘movement performance’? Was there a scale provided? How were each of these outcomes defined?
Author response: In addition to the open-structured questionnaire, we also provide the 5-point of Likert scale (5-point, strongly agree; 4-point, agree; 3-point, neither agree nor disagree; 2-point, disagree; 1-point, strongly disagree) for teachers. All teachers gave all participants a score of 3 or higher. This means positive feedback for all participants.
We provide a selection of records for reviewer.
-Diary entry, teacher interviews and Video recordings- in the abstract in addition to these three methods of recording data you also state ‘observational records’? Is this the same as the diary entrys? Consistency and clarity is required throughout the document
Author response: Observational records are the same as the diary entries. We have revised (line 18-20).
-Line 161-164: This is a sweeping statement- you should provide the data or refer to it in a past/future publication or remove these lines. Why is the data available for seven participants if only five completed the intervention?
Author response: For a more complete presentation of the results, the results of the step frequency per minute before and after step training are showing in Figure 3.
-Detail must be added to the methods section outlining how each of the outcome measures were collected e.g. in the results section you refer to body posture analysis- how would a blinded researcher replicate these assessments? The details must be provided in the methods.
Author response: Each step training session was observed and photographed by a senior physical education teacher with special education qualifications (line 113-114). Actually, the standing posture was recorded by the photographed and the two cameras. The major revised sentences highlighted in blue.
-It may also be helpful to include how you defined mild, moderate and heavy disability within the methods section.
Author response: We have added (line 95-98).
-Throughout the results section you must provide the figures from the statistical analysis in the text- you state statistical significance was found e.g. significantly greater- provide the p values etc at on these occasions.
Author response: For a more complete presentation of the results, the results of the step frequency per minute before and after step training are showing in Figure 3.
-Line 86/7: had students reached volatile exhaustion after 1 min? Did they choose to stop or did the assessor stop them? More detail required
Author response: The participants’ perceptions of the exercise intensity, which ranged from fairly light to somewhat difficult, were evaluated using Borg's Rating of Perceived Exertion (RPE) scale [23]. We have revised the step training protocol (line 101-104, 105-109).
-Line 184- the effects on physical performance – I am unclear as to how ‘physical performance’ is assessed. Firstly, your terminology around physical fitness/performance etc must be refined and consistent as discussed previously. Are you simply measuring time or duration of stepping? This is very different to measuring an aspect of physical fitness such as aerobic capacity? Please refine and define this throughout the document
Author response: Performance of daily activities, which is more indicative of our original meaning. We have modified the words and original meaning of Q1 (line 205, 213-220).
-Line 187-191- I am unsure if this is appropriate here? Reads like an opinion?
Author response: We have removed it.
-Figures and tables should be provided where they are first mentioned in text and consistent with the publishers guidelines
Author response: We have revised.
-Line 278- ‘frequency of climbing’ is introduced as another outcome measure. This is difficult to follow, you must be consistent when presenting your outcome measures and their results. You also mention physiological and psychological changes as outcome measures above here for the first time??
Author response: We have revised it (line 265-266)……obvious improvements in their walking ability, standing posture, frequency of traversing stairs, level of motivation, peer relationships, self-confidence, and ability to engage in everyday activities after completing the step training program (line 265-266).
-Line 279-281 requires a reference
Author response: We have removed these sentences in the revised manuscript.
-Check font size and colour throughout the document- there are inconsistencies
Author response: we have revised.
-Line 279-301- this section reads like a list of facts? What is this adding to the discussion of your research findings? It seems as if it may have been copied and pasted from a literature review of the area- I am unsure if it fits here. Would using the WHO physical activity guidelines and providing some statistics on participation of individuals with disability add more to this section and allow the discussion to flow into the next paragraph?
Author response: We have removed these sentences and rephrased the paragraph of discussion (line 261-266). And add WHO physical activity guideline in the discussion section of the revised manuscript (line 267-274).
-Line 313-14: Especially in adolescence, the lack of proper 313 physical stimulation may inhibit the normal development of organs and tissues …requires a reference
Author response: We have added the reference.
- Harold, W. Kohl III.; Heather, D. Cook, Editors; Physical activity and physical education: Relationship to growth, development, and health. In Educating the student body: Taking physical activity and physical education to school. Committee on Physical Activity and Physical Education in the School Environment; Food and Nutrition Board; Institute of Medicine. Washington (DC): National Academies Press (US); 2013 Oct 30.
-Line 338-44 requires references
Author response: We have added the reference.
- Hsieh, K.; Hilgenkamp, T.; Murthy, S.; Heller, T.; & Rimmer, J. H. Low levels of physical activity and sedentary behavior in adults with intellectual disabilities. International journal of environmental research and public health 2017, 14, 1503. https://doi.org/10.3390/ijerph14121503
-Line 348/9: You state… demonstrated improvement in their anxiety behavior, interpersonal relationships, and self-confidence after 8 weeks of the step training program…how can you conclude this? Make conclusions based on what your outcome measures where and your findings…for example how did you assess participant self-confidence? You subjectively asked teachers on their opinion- your conclusions must reflect your findings.
Author response: We have minor revised the sentence (line 272-274). The similar sentence has described in line 264-266.
Interviews with teachers and observations of peer interactions among students revealed that the participating students exhibited obvious improvements in their walking ability, frequency of traversing stairs, level of motivation, peer relationships, self-confidence, and ability to engage in everyday activities after completing the step training program (line 264-266).
-Line 351/55: this needs rewording.
Author response: We have removed these sentences.
-Line 358- in your conclusions you mention music tracks for the first time- this detail must be provided in your methods section.
Author response: After 6 weeks, participants began performing 8 minutes of consecutive stepping to the rhythm of lively dancing music (line 106-107). We have revised it in the material and methods section (line 106-107).
-Lines 359-61: This is the first time that encouragement from peers was discussed- this should be in your methods section.
Author response: We have revised it and added it to the material and methods section (line 110-111).
The students not participating in the program engaged in warmup and cooldown sessions and were in attendance to encourage the participants with cheers of support (line 110-11).
-There should be no new information in your conclusion section- your conclusions should simply conclude and summarise your studies findings. The conclusion section should be revised.
Author response: We have revised the abstract section (line 13-29) and conclusion section (329-333).
-Limitations of the study are not presented. This should be completed.
Author response: We have added the limitation of the study in the revised manuscript (line 321-325).
-Future research is not presented or explored. This should be provided.
Author response: We have added the future study in the revised manuscript (line 325-328).
-Line 324-26- is there an argument for using walking to music within this population? It is simple, cost effective, accessible etc- music provides motivation and studies have shown that it allows regulation of cadence to tempo. The additional aspect of walking in nature could add another potential for future research and be in line with WHO and ISPAH physical activity targets
Author response: In a safe, familiar environment, we think the music can also provide motivation for individual with intellectual disabilities to exercise. In this study, we added relaxing dance music after participants were able to perform consecutive 8-minute aerobic stepping. The type of music is decided after discussion with the participants. We also have revised the material and methods section.
We agree reviewer’s point. Music selection helps turn exercise from a chore into a fun.

Reviewer 2 Report
While the experiment appears to have been well-conducted, it is unclear what new data the study adds to the current literature. The introduction provides only a brief history, and it is unclear what 'gap' this study addresses.
- The citation method does not comply with the journal standards. References must be numbered in order of appearance in the text (including citations in tables and legends) and listed individually at the end of the manuscript. In the text, reference numbers should be placed in square brackets [ ] and placed before the punctuation; for example [1], [1–3] or [1,3]. For embedded citations in the text with pagination, use both parentheses and brackets to indicate the reference number and page numbers; for example [5] (p. 10), or [6] (pp. 101–105). Please see the Instructions for Authors section.
- Please rephrase the phrase similar studies on those with intellectual disability are few (lines 47-48). In a simple search, I found many studies, which I attach below, with the proposal that they are cited in the study:
- Lee K, Cascella M, Marwaha R. Intellectual Disability. [Updated 2021 Aug 11]. In: StatPearls [Internet]. Treasure Island (FL): StatPearls Publishing; 2021 Jan-. Available from: https://www.ncbi.nlm.nih.gov/books/NBK547654/
- Plasschaert E, Van Eylen L, Descheemaeker MJ, Noens I, Legius E, Steyaert J. Executive functioning deficits in children with neurofibromatosis type 1: The influence of intellectual and social functioning. Am J Med Genet B Neuropsychiatr Genet. 2016;171B(3):348-362.
- Li S, Tong G. An etiological study of intellectually disabled children under 14 years old in Anhui Province, China. Am J Transl Res. 2021;13(4):2670-2677. Published 2021 Apr 15.
- Au PYB, You J, Caluseriu O, et al. GeneMatcher aids in the identification of a new malformation syndrome with intellectual disability, unique facial dysmorphisms, and skeletal and connective tissue abnormalities caused by de novo variants in HNRNPK. Hum Mutat. 2015;36(10):1009-1014. doi:10.1002/humu.22837
- Fu L, Liu Y, Chen Y, Yuan Y, Wei W. Mutations in the PIGW gene associated with hyperphosphatasia and mental retardation syndrome: a case report. BMC Pediatr. 2019;19(1):68. Published 2019 Feb 27. doi:10.1186/s12887-019-1440-8
- Vissers LE, Gilissen C, Veltman JA. Genetic studies in intellectual disability and related disorders. Nat Rev Genet. 2016;17(1):9-18. doi:10.1038/nrg3999
- Patel DR, Cabral MD, Ho A, Merrick J. A clinical primer on intellectual disability. Transl Pediatr. 2020;9(Suppl 1):S23-S35. doi:10.21037/tp.2020.02.02
- Patel DR, Apple R, Kanungo S, et al. Intellectual disability: definitions, evaluation and principles of treatment. Pediatric Medicine2018;1:11 10.21037/pm.2018.12.02
- Eisenhower AS, Baker BL, Blacher J. Preschool children with intellectual disability: syndrome specificity, behaviour problems, and maternal well-being. J Intellect Disabil Res. 2005;49(Pt 9):657-671. doi:10.1111/j.1365-2788.2005.00699.x
- Balasubramanian B, V Bhatt C, A Goyel N. Genetic studies in children with intellectual disability and autistic spectrum of disorders. Indian J Hum Genet. 2009;15(3):103-107. doi:10.4103/0971-6866.60185
- Please substantiate much better the part with teachers should consider how to enhance the essential physical ability and cite studies that have addressed this issue. Same with the curriculum adaptation part. Lines 74-76.
- In addition to what is found in the text (Lines 88-90), please bring many more arguments regarding the novelties that this study brings.
- Please explain much better the criteria of inclusion/exclusion of the subjects in the study.
- Please scale the intensity part of the effort. What does mean low to medium intensity? Readers of this journal need to understand precisely how effort is classified. Lines 112-113.
- You made the following statement: The students went from being tired after 1 minute to being able to sustain the exercise for 8 minutes (Lines 186-187). This statement is very subjective and must be covered by specific "objective measurements".
- Authors should discuss and highlight the physical activity part for individuals with intellectual disabilities, including the introductory part, not just the discussions.
- On the discussion side, the authors should insist much more on the results obtained by them.
- I am inquisitive about a particular aspect that could enhance the results of this study; Did the authors also think about a Kinematic analysis? Szabo, D.A., Neagu, N., & Sopa, I. S. (2020). Kinematic angular analysis of cinematic biomechanics in forearm flexion: a case study. Geosport for Society, 13(2), 140-148. https://doi.org/10.30892/gss.1305-065
Author Response
Response to Reviewer 2
While the experiment appears to have been well-conducted, it is unclear what new data the study adds to the current literature. The introduction provides only a brief history, and it is unclear what 'gap' this study addresses.
Author response: We have carefully completed the revisions based on the reviewers' comments. The revised words or sentences are highlighted in blue. Our response to the reviewers' comments is as follows.
- The citation method does not comply with the journal standards. References must be numbered in order of appearance in the text (including citations in tables and legends) and listed individually at the end of the manuscript. In the text, reference numbers should be placed in square brackets [ ] and placed before the punctuation; for example [1], [1–3] or [1,3]. For embedded citations in the text with pagination, use both parentheses and brackets to indicate the reference number and page numbers; for example [5] (p. 10), or [6] (pp. 101–105). Please see the Instructions for Authors section.
Author response: We have revised the citation method of the references.
- Please rephrase the phrase similar studies on those with intellectual disability are few(lines 47-48). In a simple search, I found many studies, which I attach below, with the proposal that they are cited in the study:
- Lee K, Cascella M, Marwaha R. Intellectual Disability. [Updated 2021 Aug 11]. In: StatPearls [Internet]. Treasure Island (FL): StatPearls Publishing; 2021 Jan-. Available from: https://www.ncbi.nlm.nih.gov/books/NBK547654/
- Plasschaert E, Van Eylen L, Descheemaeker MJ, Noens I, Legius E, Steyaert J. Executive functioning deficits in children with neurofibromatosis type 1: The influence of intellectual and social functioning. Am J Med Genet B Neuropsychiatr Genet. 2016;171B(3):348-362.
- Li S, Tong G. An etiological study of intellectually disabled children under 14 years old in Anhui Province, China. Am J Transl Res. 2021;13(4):2670-2677. Published 2021 Apr 15.
- Au PYB, You J, Caluseriu O, et al. GeneMatcher aids in the identification of a new malformation syndrome with intellectual disability, unique facial dysmorphisms, and skeletal and connective tissue abnormalities caused by de novo variants in HNRNPK. Hum Mutat. 2015;36(10):1009-1014. doi:10.1002/humu.22837
- Fu L, Liu Y, Chen Y, Yuan Y, Wei W. Mutations in the PIGW gene associated with hyperphosphatasia and mental retardation syndrome: a case report. BMC Pediatr. 2019;19(1):68. Published 2019 Feb 27. doi:10.1186/s12887-019-1440-8
- Vissers LE, Gilissen C, Veltman JA. Genetic studies in intellectual disability and related disorders. Nat Rev Genet. 2016;17(1):9-18. doi:10.1038/nrg3999
- Patel DR, Cabral MD, Ho A, Merrick J. A clinical primer on intellectual disability. Transl Pediatr. 2020;9(Suppl 1):S23-S35. doi:10.21037/tp.2020.02.02
- Patel DR, Apple R, Kanungo S, et al. Intellectual disability: definitions, evaluation and principles of treatment. Pediatric Medicine2018;1:11 10.21037/pm.2018.12.02
- Eisenhower AS, Baker BL, Blacher J. Preschool children with intellectual disability: syndrome specificity, behaviour problems, and maternal well-being. J Intellect Disabil Res. 2005;49(Pt 9):657-671. doi:10.1111/j.1365-2788.2005.00699.x
- Balasubramanian B, V Bhatt C, A Goyel N. Genetic studies in children with intellectual disability and autistic spectrum of disorders. Indian J Hum Genet. 2009;15(3):103-107. doi:10.4103/0971-6866.60185
Author response: we have revised it (line 46-48).
“Although extensive research has examined the behaviors, patterns, and determinants of physical activity in children without disabilities, few studies have undertaken the crucial task of evaluating methods for improving physical activity in children with intellectual disabilities [10]”, (line 46-48).
- Please substantiate much better the part with teachers should consider how to enhance the essential physical abilityand cite studies that have addressed this issue. Same with the curriculum adaptation part. Lines 74-76.
Author response: The original meaning of this sentence is not about the physical ability of the teacher, but about the need for the teacher to enhance the physical activity level, physical activity motivation, and performance of students with intellectual disabilities through curriculum adaptation.
We added the reference [14] (line 67), and revised the sentence (line 69-70).
- Bertills, K.; Granlund, M.; & Augustine, L. Inclusive teaching skills and student engagement in physical education. Frontiers in Education 2019, 4, 1-13. https://doi.org/10.3389/feduc.2019.00074
……consider how to enhance the physical activity level, physical activity motivation, and performance of students with intellectual disabilities through curriculum adaptation (line 69-70).
- In addition to what is found in the text (Lines 88-90), please bring many more arguments regarding the novelties that this study brings.
Author response: We have major revised the manuscript. The changed were marked in blue.
- Please explain much better the criteria of inclusion/exclusion of the subjects in the study.
Author response: The study enrolled individuals who did not rely on walking aids. Those with major illnesses or mobility restrictions were excluded. We have revised it in material and methods section (line 89-90).
- Please scale the intensity part of the effort. What does mean low to medium intensity? Readers of this journal need to understand precisely how effort is classified. Lines 112-113.
Author response: We have revised it in material and methods section (line 101-104, 105-109).
The protocol of the step training was based on the step-test protocol, which is based on the participants' individual conditions with some modifications [22]. The initial speed was 50–80 beats per minute (BPM), and the stepper (Model 780, Taipei, Taiwan) height was set to 15 cm (line 101-104).
During week 2, participants underwent 3-minute sessions, and the session durations were increased by 1 minute per week thereafter. After 6 weeks, participants began performing 8 minutes of consecutive stepping to the rhythm of lively dancing music. The participants’ perceptions of the exercise intensity, which ranged from fairly light to somewhat difficult, were evaluated using Borg's Rating of Perceived Exertion (RPE) scale [23] (line 105-109).
- You made the following statement: The students went from being tired after 1 minute to being able to sustain the exercise for 8 minutes(Lines 186-187). This statement is very subjective and must be covered by specific "objective measurements".
Author response: We have revised it in material and methods section (line 101-104, 105-109).
- Authors should discuss and highlight the physical activity part for individuals with intellectual disabilities, including the introductory part, not just the discussions.
Author response: We have revised the introduction and discussion section and highlighted in blue.
- On the discussion side, the authors should insist much more on the results obtained by them.
Author response: We have revised the discussion section highlighted in blue.
- I am inquisitive about a particular aspect that could enhance the results of this study; Did the authors also think about a Kinematic analysis? Szabo, D.A., Neagu, N., & Sopa, I. S. (2020). Kinematic angular analysis of cinematic biomechanics in forearm flexion: a case study. Geosport for Society, 13(2), 140-148. https://doi.org/10.30892/gss.1305-065
Author response: We thank reviewer’s suggestion. The kinematic analysis method based on the combination of multi-camera motion capture system. Two cameras were arranged for this study, so that the standard kinematic analysis could not be performed.

Round 2
Reviewer 2 Report
From my point of view, you responded to all my requests in a proper way, congratulations on your work.